# Development of High Affinity Calcitonin Analog Fragments Targeting Extracellular Domains of Calcitonin Family Receptors

**DOI:** 10.3390/biom11091364

**Published:** 2021-09-15

**Authors:** Sangmin Lee

**Affiliations:** Department of Basic Pharmaceutical Sciences, Fred Wilson School of Pharmacy, High Point University, High Point, NC 27268, USA; slee2@highpoint.edu; Tel.: +1-336-841-9415

**Keywords:** calcitonin receptor, amylin receptor, peptide hormones, drug design

## Abstract

The calcitonin and amylin receptors (CTR and AMY receptors) are the drug targets for osteoporosis and diabetes treatment, respectively. Salmon calcitonin (sCT) and pramlintide were developed as peptide drugs that activate these receptors. However, next-generation drugs with improved receptor binding profiles are desirable for more effective pharmacotherapy. The extracellular domain (ECD) of CTR was reported as the critical binding site for the C-terminal half of sCT. For the screening of high-affinity sCT analog fragments, purified CTR ECD was used for fluorescence polarization/anisotropy peptide binding assay. When three mutations (N26D, S29P, and P32HYP) were introduced to the sCT(22–32) fragment, sCT(22–32) affinity for the CTR ECD was increased by 21-fold. CTR was reported to form a complex with receptor activity-modifying protein (RAMP), and the CTR:RAMP complexes function as amylin receptors with increased binding for the peptide hormone amylin. All three types of functional AMY receptor ECDs were prepared and tested for the binding of the mutated sCT(22–32). Interestingly, the mutated sCT(22–32) also retained its high affinity for all three types of the AMY receptor ECDs. In summary, the mutated sCT(22–32) showing high affinity for CTR and AMY receptor ECDs could be considered for developing the next-generation peptide agonists.

## 1. Introduction

Calcitonin (CT) a 32 amino acid peptide hormone is secreted from thyroid glands and activates the calcitonin receptor (CTR) to control calcium homeostasis. CTR is a drug target for osteoporosis treatment. Salmon CT (sCT) was developed as a peptide drug targeting human CTR due to its higher affinity and potency than human CT [1,2,3]. Interestingly, CTR can form a complex with an accessory protein called receptor activity-modifying protein (RAMP). The CTR:RAMP complexes gain affinity for the peptide hormone amylin and are known as the amylin receptor (AMY receptor). AMY receptor activation regulates blood glucose levels by reducing food intake, inhibiting glucagon secretion, and slowing gastric emptying [4]. Apparently, the AMY receptor is a drug target for diabetes treatment and also its activation holds a potential for treating other metabolic diseases including obesity [5,6,7]. A rat amylin analog pramlintide was developed and is available in clinics to treat diabetes as co-therapy with insulin [8]. In addition, lots of effort has been focused on developing next-generation peptide drugs targeting AMY receptors as exemplified with dual amylin calcitonin receptor agonists (DACRA) and long-acting amylin/calcitonin receptor agonists [9,10,11,12,13].

CTR belongs to class B G protein-coupled receptors (GPCR) that are characterized by a transmembrane domain (TM) and a large extracellular domain (ECD). The N-terminal half of CT binds CTR TM triggering G protein association and initiating cell signaling [14]. The CTR ECD is an important peptide binding site for the C-terminal half of sCT as reported in crystal structures [15,16]. Several studies have suggested the mechanisms of peptide interactions with CTR [14,15,17,18,19]. The cryo-EM structure of CTR indicated hydrogen bond interactions between CTR TM residues and sCT S5 and T6. In addition, the amphipathic helix of sCT V8, L12, L16 and L19 is located towards the hydrophobic environment of CTR TM residues [14]. Regarding CTR ECD interactions, Lee et al. used alanine-scanning mutagenesis of an sCT fragment and predicted hydrogen bond interaction between sCT T25 and the CTR ECD D101 [18]. They also reported that sCT T27, G28 and P32 were the critical residues for CTR ECD binding [18]. These predictions were confirmed with the crystal structures of the CTR ECD [15,16]. Despite the recent achievement of unveiling CTR structures, the development of next-generation peptide agonists with improved affinity and potency for CTR and AMY receptors has been limited.

The current study focused on the CTR ECD and its interaction with antagonistic sCT fragments seeking peptide ligands with higher affinity than wild-type sCT. Based on the systemic report on developing high-affinity calcitonin gene-related peptide (CGRP) analogs, I found that sCT(22–32) with N26D and S29P mutations showed a significant affinity increase for the CTR ECD compared to wild-type sCT(22–32). In addition, the crystal structures of the CTR ECD guided the introduction of hydroxyproline (HYP) to sCT(22–32) P32 for additional interaction with the CTR ECD. I also investigated the effects of this mutation on sCT(22–32) affinity. Finally, I found that sCT(22–32) with all of the N26D, S29P and P32HYP mutations showed a marked affinity increase both for the CTR ECD and the AMY receptor ECDs compared to wild-type sCT(22–32).

## 2. Materials and Methods

### 2.1. Reagents

Dulbecco’s Modified Eagle Medium (DMEM) with 4.5 g/L glucose, l-glutamine, and sodium pyruvate was obtained from Corning (Mediatech, Inc., Manassas, VA, USA) for HEK293T and HEK239S GnTI^−^ mammalian cell culture. The mixture of non-essential amino acids (NEAA, 100X) was purchased from Lonza (Basel, Switzerland). Fetal bovine serum (Cat.# F2442) was purchased from Sigma-Aldrich (St. Louis, MO, USA). DNA assembly master mix and restriction enzymes used for DNA cloning were purchased from New England Biolabs (Ipswich, MA, USA). All other reagents were purchased from Sigma-Aldrich, unless otherwise noted.

### 2.2. Cell Lines Used

HEK293T cells were purchased from ATCC (Manassas, VA, USA) for the expression of CTR ECD, RAMP1-CTR ECD fusion, and RAMP2-CTR ECD fusion proteins. HEK293S GnTI^−^ cells were also purchased from ATCC (Manassas, VA, USA) for RAMP3-CTR ECD fusion protein expression.

### 2.3. DNA Plasmids for Receptor ECD Expression

pHLsec-based vectors were used to make receptor ECD proteins secreted to cell culture media [20]. DNA assembly master mix (NEB) was used to construct DNA plasmids according to the manufacturer’s instruction. The following DNA plasmids used in this study were previously reported [21]: pHLsec/hCTR.34–141-H_6_ (H-pSL003) and pHLsec/hRAMP2.55–140-(GSA)_3_-hCTR.34–141-H_6_ (H-pSL006). The following constructs were generated for the current study: pHLsec/hRAMP1.24–111-(GSA)_3_-hCTR.34–141-H_6_ (H-pSL005) and pHLsec/hRAMP3.25–112-(GSA)_3_-hCTR.34–141-H_6_ (H-pSL001). Coding sequences of the DNA expression vectors were confirmed with Sanger sequencing performed by Psomagen (Rockville, MD, USA). DNA plasmids were purified with NucleoBond^®^ Extra Midi Plus kit (Macherey-Nagel, Dueren, Germany) from bacterial cell culture and they were stored at −20 °C until their use for transfection.

### 2.4. Expression and Purification of CTR ECD, RAMP1-CTR ECD Fusion, and RAMP2-CTR ECD Fusion Proteins

The general procedures of the receptor ECD expression from HEK293T cells were previously described [18]. Briefly, HEK293T cells transiently transfected with DNA expression vectors were incubated for 4 days at 37 °C. Cell culture media were collected and they proceeded to purification steps. All protein purification was performed at 4 °C unless otherwise noted. The cell culture media were initially dialyzed to the dialysis buffer and were followed by immobilized metal affinity column (IMAC) chromatography and size exclusion column (SEC) chromatography. The procedures of these column chromatography used in this study were previously described [18]. The final fractions from SEC chromatography containing purified receptor ECDs were dialyzed to storage buffer and stored at −80 °C until their use.

### 2.5. Expression and Purification of the RAMP3-CTR ECD Fusion Protein

HEK293S GnTI^−^ cells were used to express the RAMP3-CTR ECD fusion protein. The general procedures of receptor ECD expression and purification from HEK293S GnTI^−^ cells were previously described [22]. HEK293S GnTI^−^ cells were transiently transfected with the DNA expression vector (H-pSL001) using polyethylenimine (PEI) at 1:1.5 ratio (DNA:PEI, *w*/*w*). Transfected cells were incubated for 4 days at 37 °C. All protein purification was performed at 4 °C unless otherwise noted. Cell culture media were collected and dialyzed to dialysis buffer overnight. The next day, dialyzed cell culture media were loaded to IMAC chromatography. Peak fractions from IMAC chromatography were spin-concentrated (MWCO 10 kDa) and injected into SEC. Peak fractions from SEC chromatography were collected (Appendix A) and the purified RAMP3-CTR ECD fusion protein was confirmed with SDS-PAGE (Appendix A). When N-glycans of the RAMP3-CTR ECD fusion protein were removed by PNGase F treatment, the band of the RAMP3-CTR ECD fusion protein located closely to 25 kDa, its expected MW without any N-glycans (Appendix A). The purified RAMP3-CTR ECD fusion protein showed a selective binding profile with an antagonistic amylin analog AC413 compared to CTR ECD alone suggesting that the purified RAMP3-CTR ECD fusion protein showed amylin receptor phenotype (Appendix A). The purified RAMP3-CTR ECD fractions were dialyzed to storage buffer and stored at −80 °C until their use for peptide binding assay.

### 2.6. Synthetic Peptides

All peptides used in this study were custom-synthesized and HPLC-purified by Genscript (Piscataway, NJ, USA). An automated peptide synthesizer was used with their proprietary PepPower^TM^ peptide synthesis technology. HPLC purity of synthesized peptides was at least 85%. Mass spectrometry was performed at Genscript to confirm the molecular weight of the synthesized peptides. Fluorescein isothiocyanate (FITC)-labeled sCT(22–32), FITC-labeled AC413(6–25), and FITC-labeled AC413(6–25) with Y25P mutation were used as peptide probes for peptide binding assay. The extinction coefficient of FITC (63,000 M^−1^·cm^−1^ at 495 nm, pH 7.0) was used to determine the concentration of the FITC-labeled peptide probes. Extinction coefficients of Trp and Tyr were used to calculate the concentration of sCT(22–32) analogs. The peptide sequences used in the current study were shown in Table 1.

### 2.7. Fluorescence Polarization/Anisotropy (FP) Peptide Binding Assay

The overall procedures of FP peptide binding assay with receptor ECDs and FITC-labeled peptide probes were previously described [22]. FITC-labeled sCT(22–32) (10 nM) was used to evaluate peptide ligand affinity for the CTR ECD. AC413 has been reported as an antagonistic amylin peptide analog [23] and FITC-labeled AC413(6–25) with Y25P mutation (10 nM) was used for FP assay with AMY receptor ECDs since the Y25P mutation dramatically increased the AC413(6–25) affinity for AMY receptor ECDs [18]. A SpectraiD5 (Molecular Devices, San Jose, CA, USA) was used to measure fluorescence polarization/anisotropy. Background (reaction buffer only) was subtracted for anisotropy calculation. For the SpectraiD5, G factor (0.38 for FITC-sCT(22–32) and 0.41 for FITC-AC413(6–25) and FITC-AC413(6–25) Y25P) was used to correct the instrumental bias for anisotropy calculation. The polarization (mP) of the FITC-peptide probes only (No receptor ECD) was set close to 50 mP. For the saturation binding assay, the anisotropy values were re-calculated when total fluorescence intensity was changed by more than 10% as previously described [22]. However, effects of fluorescence intensity changes on the affinity values (K_I_) obtained from the competition binding assay was reported to be minimum [24] and re-calculation of anisotropy was not applied. For the competition binding assay, the receptor ECD concentrations (CTR ECD 500 nM, RAMP1-CTR ECD fusion protein 77 nM, RAMP2-CTR ECD fusion protein 67 nM, and RAMP3-CTR ECD fusion protein 252 nM) that produced a half of the maximal anisotropy values with the respective peptide probes were used. Anisotropy values were used to produce non-linear regression curves with PRISM 5.0 (GraphPad software, San Diego, CA, USA) as previously described [22]. K_I_ indicates affinity values obtained from competition binding assay and it was calculated from the non-linear regression curves of anisotropy by using equations previously described [22,24]. Concentrations of the FITC-labeled peptide and the total receptor protein and the affinity of the FITC-labeled peptide for the receptor protein were incorporated into the equations for the K_I_ calculation of the competitive peptide ligand. Two technical replicates were used for each receptor concentration in the peptide-binding curve. SEM of the anisotropy values of the two replicates at each receptor concentration were presented as error bars in the representative peptide-binding curves. When the error bars were shorter than the height of the symbol, they were omitted in the representative curves. At least three independent peptide binding experiments were performed to obtain three independent peptide-binding curves. Mean and standard deviation of peptide ligand affinity were calculated from them.

### 2.8. Building Hypothetical Structures

Hypothetical structures were represented by using Pymol (Schrodinger, New York, NY, USA). The crystal structure of N-glycosylated CTR ECD with sCT(16–32) peptide (PDB 6PFO, Mol A) was used for figure representation. Selective mutations were introduced to the sCT(22–32) structure by using the mutagenesis function in Pymol. Hypothetical structures of the CTR ECD complexes with RAMP1/2/3 ECD were generated by placing RAMP1/2/3 ECD structures reported in the cryo-EM structures of CGRP and AM1/2 receptors as follows. CLR ECD structures of the CGRP receptor (PDB 6E3Y) and AM1 (PDB 6UUN) and AM2 receptors (PDB 6UVA) were aligned with the CTR ECD (PDB 6PFO, Mol A). Then, CLR ECD and other domains were removed and only RAMP ECD structures were shown with the CTR ECD to build the hypothetical structures of the RAMP-CTR ECD fusion proteins. Likewise, crystal structures of RAMP1-CLR ECD fusion (PDB 4RWG) and RAMP2-CLR ECD fusion (PDB 4RWF) proteins were used to build hypothetical AMY receptor 1/2 ECD structures by placing RAMP1/2 ECD next to the CTR ECD. These hypothetical structures were represented in Appendix A. For the hypothetical structure of the CTR ECD complex with RAMP3 ECD, the additional cryo-EM structure of the AM2 receptor where AM peptide bound (PDB 6UUS) was used and the structure was shown in Appendix A. For hydroxyproline mutations, a plugin PyTMs was installed in Pymol [25] and proline hydroxylation was used to make sCT S29HYP and P32HYP mutations.

### 2.9. Statistical Analysis

PRISM 5.0 (GraphPad software, San Diego, CA, USA) was used for one-way ANOVA and Tukey’s *post hoc* test (unless otherwise noted), when more than two groups were compared. PRISM 5.0 was also used for Student’s *t*-test (two-tailed) for the statistical analysis of two groups. *p* < 0.05 was considered as a statistical significance.

## 3. Results

### 3.1. N26D/S29P Mutations of sCT(22–32) Markedly Increased Affinity for CTR ECD Compared to Wild-Type sCT(22–32)

Rist et al. used a systemic approach to address the binding sites of CGRP for the CGRP receptor [26]. They reported a high-affinity ligand for the CGRP receptor that corresponded to human alpha CGRP(27–37) with N31D, S34P, and K35F mutations (CGRPmut). Sequence alignment of CGRP(27–37), CGRPmut, and sCT(22–32) was shown in Figure 1a. Since the crystal structures of receptor ECD-bound CGRPmut appeared similar to the CTR ECD-bound sCT structure (Figure 1b), those mutations were applied to the corresponding amino acids of sCT. However, K35F mutation of CGRPmut was not applied to the corresponding sCT G30 since Lee et al. reported that the alanine mutation of sCT G30 significantly decreased the sCT(22–32) binding for the CTR ECD suggesting that the flexibility of G30 helped sCT(22–32) binding [18]. Nevertheless, when the hypothetical structure of sCT(22–32) G30F was built, the G30F mutation visually did not make a clash with the nearby CTR ECD residue N124 (Appendix A).

Figure 1c showed the hypothetical structure of sCT(22–32) with N26D/S29P mutations bound to CTR ECD. N26D and S29P mutations did not generate visual clashes with CTR ECD residues. While sCT N26D mutation did not have additional interaction with CTR ECD residues, S29P mutation appeared to fit in with the surrounding CTR ECD residue H121 and E123 (Figure 1c). The effects of these mutations on sCT(22–32) affinity were shown in Figure 1d. sCT(22–32) N26D mutation increased affinity for the CTR ECD by less than 2-fold, whereas sCT(22–32) S29P mutation increased affinity for the CTR ECD by 5-fold compared to wild-type sCT(22–32) (Figure 1d and Table 2). sCT(22–32) with N26D/S29P mutations increased affinity for the CTR ECD by 6-fold compared to wild-type sCT(22–32) (Figure 1d and Table 2) suggesting that S29P mutation was the main affinity enhancer of the sCT(22–32) fragment.

### 3.2. Mutational Effects of the sCT S29 Residue on sCT(22–32) Affinity

An sCT S29 residue was further investigated by introducing other mutations. N26D mutation increased sCT(22–32) affinity by 1.7-fold and sCT(22–32) with N26D mutation was used as a backbone peptide for the mutagenesis of the S29 residue. On the sCT(22–32) N26D backbone peptide, S29 was mutated to other amino acids with hydrophobic (S29L), hydrophobic/bulky (S29F and S29W), or charged (S29D, S29E, and S29H) side chains. Mutations to the amino acids with no or small side chains (S29G, S29A, S29V, and S29T) were also tested. Peptide analogs with each mutation at the S29 position were used at 3 μM for the competition peptide binding assay (Figure 2a). More decreased anisotropy indicated more competition by 3 μM of peptide analogs and this suggested that the peptide analogs could have a higher binding affinity for the CTR ECD than the sCT(22–32) N26D backbone peptide. Interestingly, the mutation to amino acids with small side chains at the S29 position (N26D/S29A and N26D/S29V) showed a decrease in anisotropy more than the anisotropy decrease mediated by sCT(22–32) with N26D mutation (Figure 2a). sCT(22–32) with N26D/S29P mutations showed the lowest anisotropy (Figure 2a). When multiple concentrations of the competitive peptide were used for the full binding curve, sCT(22–32) with N26D/S29A and N26D/S29V mutations showed a moderate increase in affinity compared to the sCT(22–32) N26D backbone peptide by 2.2-fold and 1.7-fold, respectively (Figure 2b and Table 2). Hypothetical structures of sCT(22–32) with N26D/S29A and N26D/S29V mutations suggested that small side chains of A29 and V29 were placed in the pocket of CTR ECD H121, E123 and N124 residues (Figure 2c). These results suggest that small and hydrophobic side chains of A29, V29, and P29 provide more favorable binding to the surrounding CTR ECD residues than a polar side chain of S29.

### 3.3. Effects of sCT S29 to Hydroxyproline Mutation on sCT(22–32) Affinity

Hydroxyproline (HYP) is an uncommon amino acid produced from proline during post-translational protein modification through hydroxylation [27]. To confirm that small and hydrophobic side chains at the sCT S29 position are more suitable for CTR ECD binding, an HYP mutation was introduced to the sCT S29 position. When sCT S29 was mutated to HYP, the hypothetical structure suggested that its hydroxyl group located toward the E123 of the CTR ECD with a 1.0 Å distance producing a visual overlap between HYP and the CTR E123 residue (Figure 3a). Consistent with the hypothetical structure, sCT(22–32) with N26D/S29HYP mutations showed a marked affinity decrease for the CTR ECD by 13-fold compared to the affinity of sCT(22–32) with N26D/S29P mutations (Figure 3b and Table 2). These results are consistent with the idea that small and hydrophobic residues at the sCT S29 position are more suitable for CTR ECD binding.

### 3.4. Hydroxyproline Mutation of sCT C-Terminal Residue P32 and Its Effects on sCT(22–32) Affinity

The C-terminal residue of sCT is proline (P32) that was reported to be critical for CTR ECD binding [15,18]. Crystal structures of CTR ECD with sCT fragments indicated that sCT P32 interacted with CTR ECD W79 [15,16]. In addition, the W79A mutation of the CTR ECD markedly decreased the potency of hCT for CTR activation supporting the critical interaction between CTR W79 and hCT P32 [17]. Since HYP has an additional hydroxyl group compared to proline and this hydroxyl group may interact with CTR ECD residues, the effects of the sCT P32 to HYP mutation on sCT(22–32) affinity were examined. The hypothetical structure of sCT(22–32) with N26D/S29P/P32HYP mutations predicted that the hydroxyl group of HYP at the sCT P32 position located in the proximity of the main chain of CTR ECD D77 with a 3.3Å distance (Figure 4a). Consistent with its potential interaction with the CTR ECD D77 main chain, the P32HYP mutation moderately increased sCT(22–32) affinity by 2.3-fold (Figure 4b and Table 2). When the P32HYP mutation was combined with the N26D/S29P mutations, the triple mutations dramatically increased sCT(22–32) affinity for the CTR ECD by 21-fold (Figure 4b and Table 2). These results indicated that the mutational effects of N26D/S29P/P32HYP were additive in sCT(22–32) affinity enhancement.

### 3.5. Mutational Effects of sCT P32HYP on sCT(22–32) Affinity for AMY Receptor ECDs

sCT is also known as a dual agonist for CTR and AMY receptors. sCT has been used to activate AMY receptors in pre-clinical research areas [28,29,30] since it displayed high binding affinity and potency for AMY receptors and showed no discrimination between CTR and AMY receptors [1,2,3]. I investigated whether the mutational effects of P32HYP, N26D/S29P, and combined N26D/S29P/P32HYP mutation(s) are conserved for AMY receptor ECDs. AMY receptor 1 and 2 ECDs were previously reported as an ECD fusion protein of RAMP1/2 ECD and CTR ECD [18]. These ECD fusion proteins were functional AMY receptor ECDs that showed selective peptide ligand binding [18]. The current study reported for the first time to my knowledge, the successful purification of functional AMY receptor 3 ECD. The RAMP3 ECD-CTR ECD fusion protein was expressed from mammalian cells and purified (Appendix A). The fusion protein showed selective peptide ligand binding suggesting that it is a functional AMY receptor 3 ECD (Appendix A).

First, the hypothetical structures of all three AMY receptor ECDs were constructed by using the CTR ECD crystal structure (PDB 6PFO, Mol A) and recently reported cryo-EM structures of CGRP (PDB 6E3Y) and AM1/2 receptors (PDB 6UUN and 6UVA). The overview of the hypothetical AMY receptor ECD structures was shown in Figure 5a. These hypothetical structures predicted that the sCT P32HYP mutation did not make any visual clash with nearby RAMP1/2/3 ECD residues. The main chain of RAMP1 ECD W84 is located at a 4.5Å distance from the hydroxyl group of sCT HYP32 (Figure 5b). The main chain of RAMP2 ECD F111 is located at a 5.9Å distance from the hydroxyl group of sCT HYP32 (Figure 5c) and the side chain of RAMP3 ECD W84 located at a 5.2Å distance from the hydroxyl group of sCT HYP32 (Figure 5d). In addition, there are more RAMP1/2/3 ECD structures reported. When they were used to build hypothetical AMY receptor ECD structures (Appendix A), the distance from the hydroxyl group of sCT HYP32 to the RAMP ECD residues ranged from 3.9Å to 5.2Å. In these hypothetical structures, RAMP ECD residues at AMY receptor ECDs would not inhibit CTR ECD interactions with sCT HYP32. These predictions are in line with the idea that the mutational effects of P32HYP on sCT(22–32) affinity for CTR ECD will be conserved for AMY receptor ECDs.

The affinity of sCT(22–32) with P32HYP, N26D/S29P, or N26D/S29P/P32HYP mutation(s) was evaluated with AMY receptor 1/2/3 ECDs (Figure 5e–g and Table 2). Previously, AC413(6–25) with Y25P mutation (an antagonistic amylin analog) showed a relatively high affinity for AMY receptor ECDs (IC_50_ 162 nM and 76 nM for AMY receptor 1/2 ECD, respectively) [18]. Accordingly, the current study used FITC-labeled AC413(6–25) with Y25P mutation for the FP peptide binding assay. As expected, the sCT P32HYP mutation increased sCT(22–32) affinity for the AMY receptor 1/2/3 ECD by 2- to 2.9-fold. In addition, sCT N26D/S29P mutations increased sCT(22–32) affinity for the AMY receptor 1/2/3 ECDs by 6- to 8-fold. The combined N26D/S29P/P32HYP mutations markedly increased sCT(22–32) affinity for the AMY receptor 1/3 ECDs by over 20-fold. In contrast, the combined mutations increased sCT(22–32) affinity by 8-fold for the AMY receptor 2 ECD and this affinity increase was not significantly different from the affinity increase mediated by N26D/S29P mutations (Table 2). These results indicated that the affinity enhancement of sCT(22–32) mediated by P32HYP, N26D/S29P and N26D/S29P/P32HYP mutations were conserved for the AMY receptor 1/2/3/ECDs with a relatively weaker effect on the AMY receptor 2 ECD.

## 4. Discussion

This study for the first time reported the mutations of sCT that significantly enhanced sCT(22–32) affinity for the CTR ECD. The N26D and S29P mutations of sCT(22–32) markedly increased affinity for the CTR ECD by 6-fold and the additional P32HYP mutation further increased sCT(22–32) affinity by over 2-fold compared to the wild-type sCT(22–32). Accordingly, the sCT(22–32) with the combined mutations (N26D, S29P, and P32HYP) increased affinity for the CTR ECD by 21-fold compared to wild-type sCT(22–32) and this affinity enhancement was conserved for all three types of the AMY receptor ECDs. This peptide analog fragment with improved affinity for the CTR and the AMY receptor ECDs could be considered for developing next-generation peptide drugs targeting CTR and AMY receptors.

sCT has been shown to have higher affinity and potency for human CTR than hCT [1,2,3] and eel calcitonin (eCT) was also reported to show cell signaling potency similar or superior to sCT [31]. Interestingly, eCT has D26, V27, and A29 compared to sCT where it has N26, T27, and S29. One of the DACRAs KBP-042 has D26, V27, and A29 consistently with eCT [32]. This study also reported that N26D and S29A mutations (making sCT more like eCT) increased sCT(22–32) affinity for CTR ECD by 4-fold compared to wild-type sCT(22–32) (Table 2, Figure 1d and Figure 2b). These results are consistent with the previous reports showing that eCT and KBP-042 showed similar or superior effects to sCT [31,32]. Lee et al. previously reported that S29A mutation increased sCT(22–32) binding for the CTR ECD analyzed by beads-based A-LISA peptide binding assay [18] and this is also in line with the findings of the current study.

The mutational strategies for developing high-affinity sCT peptide fragments came from the systemic report on CGRP and its high-affinity analogs [26]. In this report, Rist et al. developed three mutations of CGRP N31D, S34P and K35F that dramatically increased CGRP(27–37) affinity for the CGRP receptor. Recent structural studies on the CGRP receptor and the CTR ECD indicated that the receptor-binding mode of CGRP(27–37) with N31D, S34P, and K35F mutations (CGRPmut) was similar to that of sCT(22–32) [15,16,33]. This supports the idea that applying affinity-enhancing mutations to the corresponding residues of sCT would increase sCT(22–32) affinity for the CTR ECD. The current study provided evidence that this was the case for sCT(22–32) and the CTR ECD. Although the CGRPmut showed a 100-fold increase in CGRP affinity for its receptor [26], sCT(22–32) with the corresponding two N26D and S29P mutations showed a 6-fold increase in sCT(22–32) affinity. Rist et al. also tested the HYP mutation at the position of CGRP S34. CGRP(27–37) with N31D, S34HYP, and K35F mutations decreased CGRP(27–37) affinity by 30-fold compared to CGRP(27–37) with N31D, S34P, and K35F (CGRPmut) [26]. This suggests that an additional hydroxyl group of HYP may create a steric clash which resulted in the dramatic decrease in affinity. This result was also consistent with the current study with sCT(22–32) with the corresponding S29HYP mutation where sCT(22–32) affinity was markedly decreased by 13-fold (Figure 3b).

The sCT(22–32) fragment has P32 as a C-terminal residue and it has been shown that P32 is critical for CTR ECD binding [15,18]. Lee et al. reported that P32A or P32Y mutation markedly decreased sCT(22–32) binding for CTR ECD [18]. Consistently, Johansson et al. reported that the P32Y mutation greatly decreased sCT(8–32) affinity for full length CTR [15]. When sCT P32 was mutated to HYP, HYP would retain the interaction with CTR W79 and also an additional hydroxyl group of HYP could provide a potential interaction site for other CTR ECD residues. Consistently, an HYP mutation at sCT P32 increased sCT(22–32) affinity moderately by 2.3-fold with a statistical significance (Figure 4b and Table 2). The current study provided evidence that introducing an HYP mutation to sCT P32 could be useful for designing peptide ligands with improved receptor binding affinity.

CTR forms a complex with RAMP and the complexes gain affinity for the peptide hormone amylin [1,2,34]. RAMP ECDs are known to provide limited access to the peptide hormones at CGRP and AM receptors and only C-terminal residues of CGRP and AM peptides were shown to contact RAMP ECDs [33]. Accordingly, the amylin C-terminal residue Y37 has been suggested as a potential site for RAMP interaction [21,35]. Amylin Y37 was reported to enhance amylin potency for AMY receptor 1/3 activation [35] and it was also reported to interact with RAMP2 residue E101 [21]. While amylin Y37 holds a hydroxyl group in its bulky side chain, sCT C-terminal residue P32 has a relatively small side chain. When the sCT P32HYP mutation introduced a hydroxyl group to P32, the additional hydroxyl group did not appear to interact with the RAMP ECD residues (Figure 5a–d). Consistently, the affinity enhancement mediated by the sCT P32HYP mutation was also conserved with the AMY receptor ECDs (Figure 5e–g). Structural information from either crystal structures of the AMY receptor ECDs or cryo-EM structures of full length AMY receptors will be essential to clearly elucidate how RAMPs interact with peptide ligands at AMY receptors.

One of the assumptions that initiated this study was that the CGRPmut and sCT fragments have a similar binding mode at their respective receptor ECDs. Booe et al. have reported extensive peptide interaction studies with CGRP and AM receptors to develop peptide analogs with improved affinity/potency profiles [36,37]. Their high-affinity CGRP and AM peptide analogs might be applicable to CTR and AMY receptors. The binding profiles of those peptide ligands for CTR and in particular for all three AMY receptor ECDs remain to be tested, although the receptor binding of those peptides would not be selective for CTR and AMY receptors.

The next important question is how to make peptide agonists with enhanced affinity and potency for CTR and AMY receptors. This study covers the sCT(22–32) fragment that binds the ECD portion of CTR and AMY receptors. These receptors are class B GPCRs and their activation model has been suggested [38,39]. The dynamic ECD of these receptors facilitates the initial binding of the peptide C-terminal part. The following interaction of the N-terminal part of the peptides with the receptor TM activates the receptors for G protein association triggering cell signaling. Booe et al. developed CGRP and AM peptide fragments with nanomolar affinity for their respective receptor ECDs [36,37]. Using these high-affinity peptide fragments, they further developed picomolar affinity antagonists targeting full-length CGRP and AM receptors [37]. Unexpectedly, when they made the agonist versions of these peptides with affinity-enhancing mutations, the potency enhancement of those agonists was not apparent as the affinity enhancement shown for the receptor ECDs [37]. However, the affinity enhancement for receptor ECDs was shown to increase peptide residence time at the CGRP and AM receptors and the agonists with affinity-enhancing mutations turned out to be long-acting agonists [37]. Whether the agonist version of sCT(22–32) with affinity-enhancing mutations increases the residence time at the CTR and whether it shows the long-acting property at the CTR are of great interest and remain to be investigated in future studies.

The efficacy of the dual agonists of CTR and AMY receptors for metabolic diseases has been largely attributed to their activity on the AMY receptor [40]. However, recent reports showed that the dual agonist KBP-088 and the combined use of amylin and CT were superior to activating either the AMY receptor or CTR alone [41,42]. These results suggest that the activation of CTR itself is involved in those metabolic processes and that CTR may be a valid player for the efficacy of the dual agonists. In addition, a long-acting amylin analog (LAAMA) was reported to decrease body weight gain in RAMP1 or RAMP3 knockout mice given a high fat diet [9]. These results indicate that CTR alone activated by LAAMA was enough to prevent body weight gain in mice. The current study provided the peptide analog that showed a higher affinity for CTR and AMY receptor ECDs than wild-type sCT(22–32). The mutations that enhanced sCT(22–32) affinity could be exploited to develop the peptide agonists with improved affinity and potency for these two important receptors.

## Figures and Tables

**Figure 1 biomolecules-11-01364-f001:**
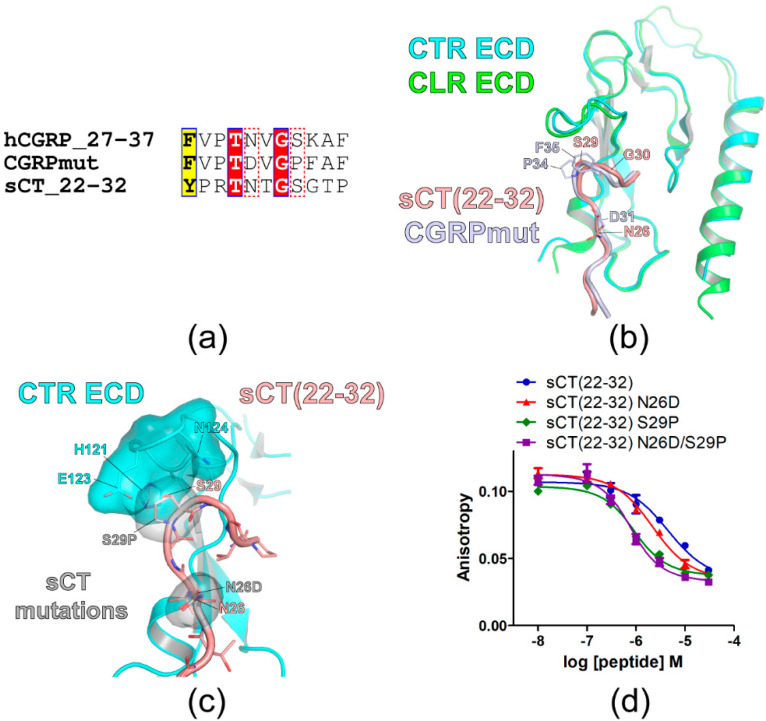
Research strategies for the introduction of sCT mutations and the mutational effects on sCT(22–32) affinity for CTR ECD. (**a**) Sequence alignment of the peptide ligands. The sequence of human α-CGRP (hCGRP) was used for hCGRP_27–37 where the number indicates the amino acid number. CGRPmut was the hCGRP_27–37 fragment with N31D, S34P, and K35F mutations. The amino acid sequence of sCT(22–32) was also shown. A similar residue was shown in black bold characters and boxed in yellow. Identical residues were shown in white bold characters and boxed in red. N31 and S34 of hCGRP_27–37 and corresponding residues of CGRPmut and sCT_22–32 were boxed with a red dotted line. (**b**) Crystal structures of CTR ECD with sCT(16–32) (PDB 6PFO, Mol A) and CLR ECD (PDB 4RWG) with CGRPmut were aligned. sCT(16–21) structure was omitted and only sCT(22–32) structure was shown. (**c**) The hypothetical structure of CTR ECD with sCT(22–32) with mutations. sCT N26D and S29P mutations were introduced and the mutated residues were superimposed with sCT wild-type residues. CTR ECD H121, E123, and N124 and sCT N26D and S29P mutated residues were shown both with stick and surface representations. (**d**) FP competition peptide binding assay with sCT(22–32) with mutations and purified CTR ECD. FITC-sCT(22–32) was used as a peptide probe. Representative peptide-binding curves were shown from three independent experiments.

**Figure 2 biomolecules-11-01364-f002:**
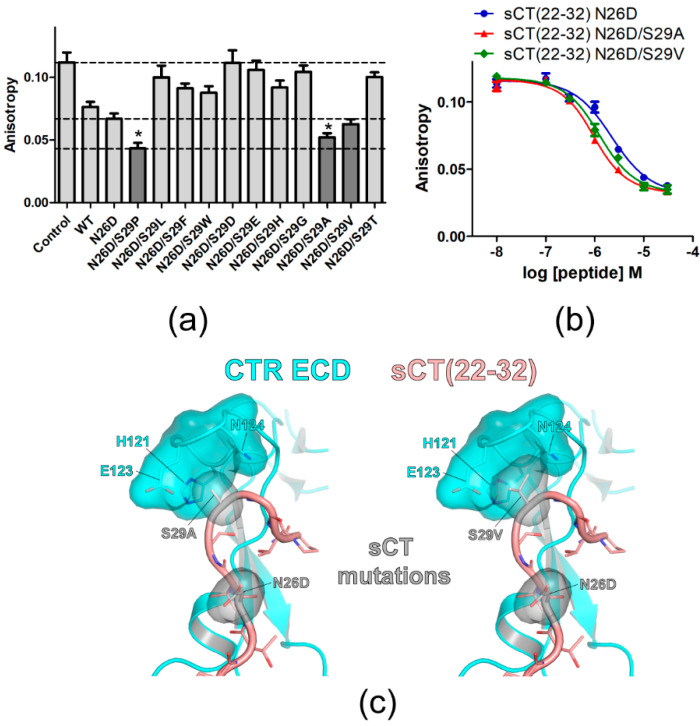
Mutational effects of sCT S29 residue on sCT(22–32) affinity. (**a**) FP competition peptide binding assay with 3 μM peptide analogs with mutations at S29. The mutations showing lower anisotropy values than that of sCT(22–32) N26D backbone peptide were indicated in dark gray. For comparison, anisotropy values of Control, sCT(22–32) with N26D mutation and sCT(22–32) with N26D/S29P mutations were indicated with dotted lines. Two technical replicates were used for each experiment. Anisotropy values from three independent experiments (each experiment has two technical replicates of anisotropy values for each peptide) were combined and the average anisotropy was shown with S.D. as error bars. Asterisk (*) indicates anisotropy decreases with a statistical significance (*p* < 0.05) compared to sCT(22–32) N26D. ANOVA with Dunnett’s *post hoc* test was performed with PRISM. sCT(22–32) N26D was used as a control group for the *post hoc* test. (**b**) FP competition peptide binding assay with sCT(22–32) N26D/S29A and N26D/S29V mutations. FITC-sCT(22–32) was used as a peptide probe. Representative peptide-binding curves were shown from three independent experiments. (**c**) Hypothetical structures of the CTR ECD (PDB 6PFO, Mol A) with sCT(22–32) with N26D/S29A (left) and N26D/S29V (right) mutations. CTR ECD H121, E123, and N124 and sCT N26D, S29A, and S29V mutated residues were shown both with stick and surface representations.

**Figure 3 biomolecules-11-01364-f003:**
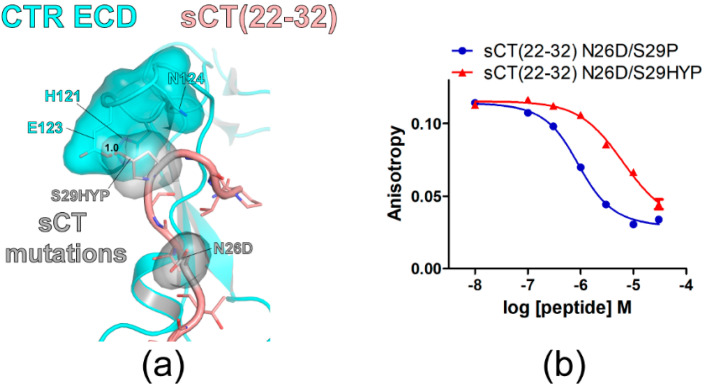
sCT S29 to hydroxyproline mutation markedly decreased sCT(22–32) affinity for CTR ECD. (**a**) The hypothetical structure of CTR ECD (PDB 6PFO, Mol A) with sCT(22–32) N26D/S29HYP mutations. CTR ECD H121, E123, and N124 and sCT N26D and S29HYP mutated residues were shown both with stick and surface representations. The distance between sCT HYP29 and CTR ECD E123 was shown as a dotted red line and was measured as 1.0 Å. (**b**) FP competition peptide binding assay with sCT(22–32) N26D/S29P and N26D/S29HYP mutations. FITC-sCT(22–32) was used as a peptide probe. Representative peptide-binding curves were shown from three independent experiments.

**Figure 4 biomolecules-11-01364-f004:**
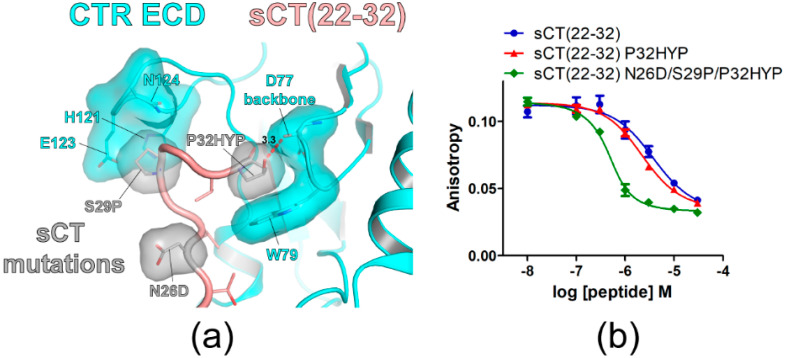
sCT P32 to hydroxyproline mutation increased sCT(22–32) affinity for CTR ECD. (**a**) The hypothetical structure of CTR ECD (PDB 6PFO, Mol A) with sCT(22–32) N26D/S29P/P32HYP mutations. CTR ECD D77, W79, H121, E123, and N124 and sCT N26D, S29P and P32HYP mutated residues were shown both with stick and surface representations. The distance between sCT HYP32 and the CTR ECD D77 main chain was shown as a dotted red line and was measured as 3.3 Å. (**b**) FP competition peptide binding assay with sCT(22–32) P32HYP and N26D/S29P/P32HYP mutations. FITC-sCT(22–32) was used as a peptide probe. Representative peptide-binding curves were shown from at least three independent experiments.

**Figure 5 biomolecules-11-01364-f005:**
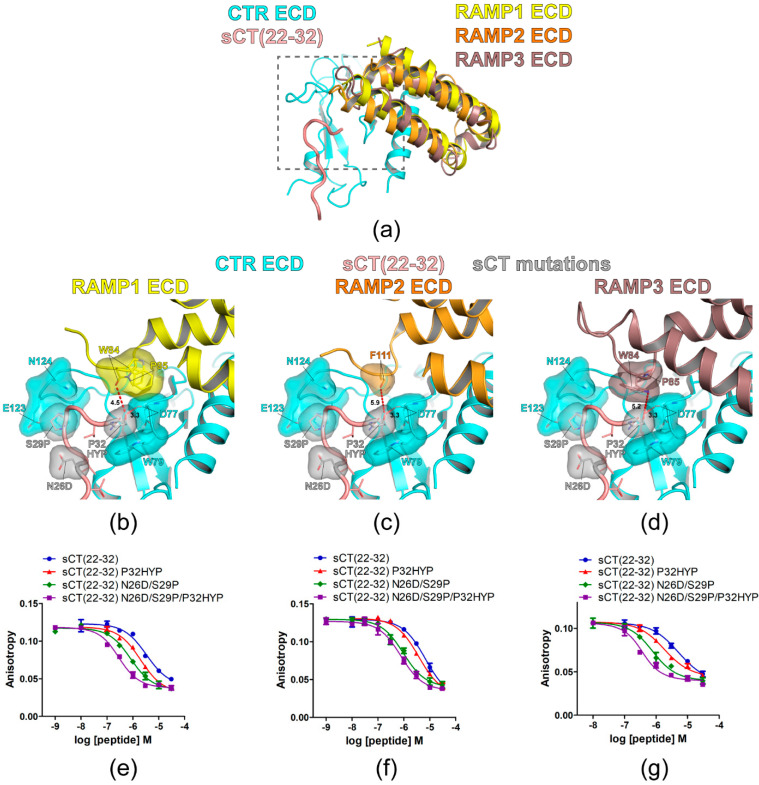
Hypothetical structures of AMY receptor ECDs and the peptide binding assay with sCT(22–32) N26D/S29P/P32HYP mutations. (**a**) Overview of the hypothetical structures of AMY receptor 1/2/3 ECDs. The crystal structure of CTR ECD (PDB 6PFO, Mol A) with sCT(22–32) and RAMP 1/2/3 ECDs structures from the CGRP (PDB 6E3Y) and AM1/2 receptors (PDB 6UUN and 6UVA) were used to build hypothetical AMY receptor 1/2/3 ECDs. The peptide binding pocket and proximal RAMP ECD residues were boxed with a gray dotted line. (**b**–**d**) Hypothetical structures of AMY receptor 1/2/3 ECDs with sCT(22–32) N26D/S29P/P32HYP mutations. CTR ECD D77, W79, H121, E123, and N124, sCT N26D, S29P, and P32HYP mutated residues, and RAMP ECD residues (W84 and P85 in RAMP1/3 ECD and F111 in RAMP2 ECD) were shown both with stick and surface representations. The distances between sCT HYP32 and the proximal RAMP ECD residue and between sCT HYP32 and the CTR ECD D77 main chain were shown as a dotted red line. For (**c**), the side chain of F111 of RAMP2 ECD was unavailable in the original cryo-EM structure (PDB 6UUN) and only the first carbon of the side chain was shown as stick and surface representations. (**e**–**g**) FP competition peptide binding assay with sCT(22–32) N26D/S29P/P32HYP mutations. FITC-AC413(6–25) with Y25 mutation was used as a peptide probe. Representative peptide-binding curves were shown from three independent experiments.

**Table 1 biomolecules-11-01364-t001:** Peptide sequences used in the current study.

Peptide Name	Peptide Sequence
FITC-sCT(22–32)	FITC-(Ahx)-YPRTNTGSGTP-NH_2_
FITC-AC413(6–25)	FITC-(Ahx)-ANFLVRLQTYPRTNVGANTY-NH_2_
FITC-AC413(6–25) Y25P	FITC-(Ahx)-ANFLVRLQTYPRTNVGANTP-NH_2_
sCT(22–32)	YPRTNTGSGTP-NH_2_
sCT(22–32) N26D	YPRTDTGSGTP-NH_2_
sCT(22–32) S29P	YPRTNTGPGTP-NH_2_
sCT(22–32) N26D/S29P	YPRTDTGPGTP-NH_2_
sCT(22–32) N26D/S29L	YPRTDTGLGTP-NH_2_
sCT(22–32) N26D/S29F	YPRTDTGFGTP-NH_2_
sCT(22–32) N26D/S29W	YPRTDTGWGTP-NH_2_
sCT(22–32) N26D/S29D	YPRTDTGDGTP-NH_2_
sCT(22–32) N26D/S29E	YPRTDTGEGTP-NH_2_
sCT(22–32) N26D/S29H	YPRTDTGHGTP-NH_2_
sCT(22–32) N26D/S29G	YPRTDTGGGTP-NH_2_
sCT(22–32) N26D/S29A	YPRTDTGAGTP-NH_2_
sCT(22–32) N26D/S29V	YPRTDTGVGTP-NH_2_
sCT(22–32) N26D/S29T	YPRTDTGTGTP-NH_2_
sCT(22–32) N26D/S29HYP	YPRTDTG[HYP]GTP-NH_2_
sCT(22–32) P32HYP	YPRTNTGSGT[HYP]-NH_2_
sCT(22–32) N26D/S29P/P32HYP	YPRTDTGPGT[HYP]-NH_2_

Mutated residues were shown in red. Ahx, aminohexanoic acid.

**Table 2 biomolecules-11-01364-t002:** The affinity of peptide analogs for CTR and AMY receptor ECDs.

Figure and Receptor ECD	Competitive sCT(22–32) Analog	N	pK_I_ Mean ± SD	Mean K_I_ (nM)
Figure 1d CTR ECD	sCT(22–32)	3	5.75 ± 0.10	1760
	sCT(22–32) N26D	3	6.00 ± 0.03 ^1^	1000
	sCT(22–32) S29P	3	6.46 ± 0.06 ^1,2^	343
	sCT(22–32) N26D/S29P	3	6.50 ± 0.13 ^1,2^	314
Figure 2b CTR ECD	sCT(22–32) N26D	3	6.01 ± 0.01	980
	sCT(22–32) N26D/S29A	3	6.36 ± 0.13 ^2^	437
	sCT(22–32) N26D/S29V	3	6.23 ± 0.11	591
Figure 3b CTR ECD	sCT(22–32) N26D/S29P	3	6.62 ± 0.15	239
	sCT(22–32) N26D/S29HYP	3	5.50 ± 0.03 ^3^	3180
Figure 4b CTR ECD	sCT(22–32)	4	5.69 ± 0.09	2030
	sCT(22–32) P32HYP	3	6.05 ± 0.04 ^1^	887
	sCT(22–32) N26D/S29P/P32HYP	3	7.02 ± 0.12 ^1,4^	95
Figure 5e AMY receptor 1 ECD	sCT(22–32)	3	5.59 ± 0.18	2550
	sCT(22–32) P32HYP	3	6.05 ± 0.03 ^1^	887
	sCT(22–32) N26D/S29P	3	6.42 ± 0.08 ^1,4^	381
	sCT(22–32) N26D/S29P/P32HYP	3	6.91 ± 0.15 ^1,3,4^	124
Figure 5f AMY receptor 2 ECD	sCT(22–32)	3	5.55 ± 0.11	2800
	sCT(22–32) P32HYP	3	5.85 ± 0.10 ^1^	1430
	sCT(22–32) N26D/S29P	3	6.33 ± 0.06 ^1,4^	470
	sCT(22–32) N26D/S29P/P32HYP	3	6.47 ± 0.02 ^1,4^	340
Figure 5g AMY receptor 3 ECD	sCT(22–32)	3	5.59 ± 0.03	2560
	sCT(22–32) P32HYP	3	6.00 ± 0.07 ^1^	995
	sCT(22–32) N26D/S29P	3	6.48 ± 0.04 ^1,4^	333
	sCT(22–32) N26D/S29P/P32HYP	3	6.93 ± 0.07 ^1,3,4^	118

^1^ *p* < 0.05 compared to sCT(22–32). ^2^ *p* < 0.05 compared to sCT(22–32) N26D. ^3^ *p* < 0.05 compared to sCT(22–32) N26D/S29P. ^4^ *p* < 0.05 compared to sCT(22–32) P32HYP. ANOVA with Tukey’s multiple comparison test was used for statistical analysis except for Figure 3b where Student’s *t*-test was used. N indicates numbers of independent experiments. pK_I_ indicates –log_10_[K_I_] (K_I_ as molar concentration).

## Data Availability

The data underlying this article will be shared on reasonable request.

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
