# Peer review of "Development of High Affinity Calcitonin Analog Fragments Targeting Extracellular Domains of Calcitonin Family Receptors"

_biomolecules, 2021, doi:10.3390/biom11091364_

Round 1

Reviewer 1 Report

Calcitonin and amylin receptors are important targets for next generation peptide drugs to improve treatments for osteoporosis and diabetes. This manuscript describes the design of peptide analogs based upon the C-terminal segment of calcitonin that bind with high affinity to the extracellular domain of the calcitonin and amylin receptors.

Overall, the manuscript is well written and adequately referenced to recent and relevant literature. However, there are a few instances of incorrect grammar and the use of personal pronouns should be reconsidered. Examples include: line 32 “are knowns”; lines 47-49 is not a complete sentence; line416 “it has been known” should be edited to “is known” or “has been shown”. Please check the references for correct capitalisation of titles as a mix of styles is currently displayed.

The methods state that two technical replicates were used (line 162) but the figures refer to three independent experiments (lines 230 and 370, for example). Please clarify.

The main concern that I have with the scientific methods is that no rigorous computational modelling has been performed, either in the design of analogues or analysis of results. Only model representations have been constructed using PyMol. Yet the results are discussed in terms of ‘steric clashes’. By what methods were these steric clashes measured or assessed? If computational modelling was not performed then terminology should be carefully selected to reflect this omission. For example, figures showing the peptides bound to CTR ECD are speculative models rather than the ‘results’ of computational modelling.

What is the scale for anisotropy in the binding assay figures?

Were statistics performed on the data of Figure 2a? For example, the results of N26D/S29V do not appear statisticaly different from N26D, and N26D/S29A only marginally so.

Reviewer 2 Report

Generally, the manuscript is well written, results are clearly presented and no major critical observations were made during the review.

Nevertheless, there are few minor comments:

1) Prior to publishing manuscript should undergo English grammar and language check, since there were some unfortunate errors such as:

  • "(...) and activates the calcitonin receptor (CTR) to controls calcium homeostasis." page 1 line 27
  • "The CTR:RAMP complexes gain affinity for peptide hormone amylin and are knowns as the amylin receptor (AMY receptor)." page 1 lines 32-33

2) Although synthetic peptides were acquired from third company, detailed description of their synthesis, quality control and purity should be implemented into the manuscript.

3) I suggest to implement the table presenting sequences of studied peptides into chapter 2.5 Synthetic peptides. This is due to the fact that those sequences are the main topic of this manuscript

4) Along the manuscript author multiple time refers to peptide analogs using the name of reference molecule. As an example:

"N26D and S29P mutations of the sCT(22-32) fragment markedly increased sCT(22-32) affinity for CTR ECD by 6-fold(...)" page 12 lines 379-380

In my opinion such statement would be true only if the sCT(22-32) N26D/S29P was an affinity enhancer of sCT(22-32) (i.e.  synergistic effect of both molecules). Otherwise, we should describe the analogue as having 6-fold higher affinity for the receptor when compared to reference molecule.

Reviewer 3 Report

The manuscript is devoted to the comparative analysis of the affinity of mutant forms of the salmon calcitonin peptide fragment (sCT(22-32)) with respect to calcitonin and amylin receptors (more exactly, extracellular domains of these receptors). Affinity comparisons were performed using the method of fluorescence polarization anisotropy and a non-standard version of the competitive binding assay. The author found mutations that significantly improved affinity of the sCT(22-32) peptide to these receptors. The results obtained are important for understanding the structural basis of the interaction of calcitonin and amylin receptors with peptide agonists. They are also of interest for an applied pharmacology. The manuscript can be recommended for publication in Biomolecules after minor revision. 

I have two essential and several technical notes.

Essential notes.

  1. Molecular modeling, which was widely used in the manuscript, usually requries molecular dynamics (MD) relaxation of the created models. MD was not performed in the peer-reviewed work. Without MD any conclusions/predictions regarding the result of point mutations are superficial.  
  2. The "competitive binding assay" used in the manuscript is not a classic one.  It was realized with 10 nM fluorescent ligand and a large excess of a receptor (500 nM CTR ECD). Addition of a competitive ligand will negligibly affect a complex between the fluorescent ligand and the receptor until the concentration of a free receptor is significantly reduced.  If so, the question arise, how was Ki=95 nM obtained for N26D/S29P/P32HYP mutant? Is there any other factor that affect an interaction of the fluorecent ligand with the receptor in the presence of the mutant? For example, formation of a complex between fluorescent ligand and mutant peptide?       

Technical notes.

1. Abstract. Line 15. Delete "markedly".

Introduction

2. Lines 30-31.  ...an accessary protein called receptor activity-modifying proteins.." You should use either the singular or plural in both cases.

3.Line 32. Change "knowns" to known.

4. Lines 53-56. Sentence "...development of sCT analogs outperforming sCT has not been available yet" should be corrected. 

5. Line 60. Better "wild-type CGRP"

6. Lines 70-71. "This study... investigated ...  and reported..." Formally, the study cannot investigate and report something. 

7. Lines 107-109. Check the gramma in the sentence "The collected cell culture..."

8. Lines 155-156. "the affinity values (Ki)". Please, explain more precisely, what is Ki and how is it measured/calculated.

Results

9. Lines 191-200. In fact, this is a repetition of the text (lines 58-66) from the Introduction. I propose to transfer the text describing a logic of the study (lines 58-66) from the introduction to the beginning of the results section and thereby to avoid repetition.  

10. Lines 209 and 210: delete "markedly" and "both", respectively. 

11. Figure 1 legend. Delete a sentence "ClustalX2 was..." or transfer it to the Material and Methods.

12. Lines 222-223. Correct the grammar: "... mutation.... was used as a backbone peptide".

13. Table 1. Please, include an explanation in the table footnote, what is N and pKi.

14. Table 1. Separate data related to different AMY receptors by horizontal lines.

15. Line 454. Correct the grammar: "...facilitates the initial binding site..."

16/ Line 467. Change "for" to "in"     
